# The Interplay between Integrins and Immune Cells as a Regulator in Cancer Immunology

**DOI:** 10.3390/ijms24076170

**Published:** 2023-03-24

**Authors:** Qingfang Zhang, Shuo Zhang, Jianrui Chen, Zhenzhen Xie

**Affiliations:** 1College of Basic Medical, Nanchang University, Nanchang 330006, China; 2Queen Mary School, Medical Department, Nanchang University, Nanchang 330031, China

**Keywords:** integrins, immune cells, cancer, immunity, cancer immunotherapy

## Abstract

Integrins are a group of heterodimers consisting of α and β subunits that mediate a variety of physiological activities of immune cells, including cell migration, adhesion, proliferation, survival, and immunotolerance. Multiple types of integrins act differently on the same immune cells, while the same integrin may exert various effects on different immune cells. In the development of cancer, integrins are involved in the regulation of cancer cell proliferation, invasion, migration, and angiogenesis; conversely, integrins promote immune cell aggregation to mediate the elimination of tumors. The important roles of integrins in cancer progression have provided valuable clues for the diagnosis and targeted treatment of cancer. Furthermore, many integrin inhibitors have been investigated in clinical trials to explore effective regimens and reduce side effects. Due to the complexity of the mechanism of integrin-mediated cancer progression, challenges remain in the research and development of cancer immunotherapies (CITs). This review enumerates the effects of integrins on four types of immune cells and the potential mechanisms involved in the progression of cancer, which will provide ideas for more optimal CIT in the future.

## 1. Introduction

Integrins, a superfamily of cell adhesion receptors, bind to extracellular matrix (ECM) ligands and cell surface ligands to mediate physiological activities [1]. Integrins are composed of a transmembrane α subunit and β subunit, with 18 α subunits and 8 β subunits currently known, constituting 24 heterodimers in humans that are divided into four categories: RGD receptors, leucocyte-specific receptors, collagen receptors, and laminin receptors [1] (Figure 1). Each heterodimer is generated from a combination of a large extracellular domain, a transmembrane domain, and a short cytoplasmic domain linked to the cytoskeleton [2]. Integrins induce inside-out and outside-in signaling pathways by transitioning from an inactive to an active state based on their structure [1]. Indeed, the binding and dissociation of the cytoplasmic domain of α and β subunits are confirmed to determine the state of integrins [1]. In the inside-out pathway, chemokines transmit signals through G protein-coupled receptors to phosphorylate the cytoplasmic domain of the β subunit [1]. Meanwhile, binding of the cytoskeletal cohesion protein talin to the cytoplasmic tail of a β subunit results in the dissociation of α and β tails and induces conformational changes in the extracellular region that increase the affinity of integrins for their ligands [1]. In the outside-in pathway, ligand binding induces conformational changes to integrins clustered on the plasma membrane and separates their heads and tails, leading to the interaction of the cytoplasmic tail domain with intracellular signaling molecules [1].

Integrins regulate various physiological or pathological processes in diverse cells [1,3]. Immune cells contain integrins that regulate cellular activity to maintain immunity and homeostasis [4,5,6,7]. However, dysregulation of integrins contributes to immune-related diseases such as multiple sclerosis (MS), arthritis, cancer, and certain diseases involving the interaction of immune cell integrins with their receptors, including Alzheimer’s and Parkinson’s diseases, pulmonary pathologies (such as asthma and chronic obstructive pulmonary disease), and atherosclerosis [8,9,10,11,12,13]. Notably, the binding of integrins to various ligands on the surface of immune cells or cancer cells promotes or inhibits the occurrence and development of cancers. Based on comprehensive analyses of numerous studies, this paper summarizes three precancerous mechanisms related to integrins: mediating the overgrowth of cancer cells, promoting invasion and metastasis of cancer cells, and generating immune tolerance effects by the inhibition of immune cell responses [14,15,16]. However, integrins also inhibit cancer through diverse mechanisms [17,18]. The pronounced correlation of integrins with underlying cancer mechanisms has led to substantial research on integrins in the field of cancer therapy. Currently, three main integrin-based therapies are in development: chimeric antigen receptor (CAR) T-cell therapy, oncolytic virotherapy, and immune checkpoint inhibitor (ICI) therapy. With the role of integrins in the development of immune cells and cancer becoming clear, more cancer therapeutic approaches will be likely to be developed to support the expansion of oncotherapy (Table 1).

## 2. Potential Influence of Integrins on Immune Cells

### 2.1. B Cells

Multiple types of integrins regulate the activities of B cells in humoral immunity, such as adhesion, migration, and proliferation. As an outside-in signaling mechanism, it has been reported that αX integrins on tonsillar B cells bound to fibrinogen induce cell proliferation, adhesion, and migration [87]. As an inside-out signaling mechanism, integrins were shown to be activated by talin-1, a key integrin-coactivator, which is critical in the migration of B cells into lymph nodes [88]. Similarly, Hart et al. indicated that αLβ2 integrins that are stimulated by the integrin-activated protein kindlin enhanced the binding of αLβ2 integrins to the intercellular adhesion molecule 1 (ICAM-1), thereby mediating the movement of B cells [89]. In addition, some experiments have confirmed the role of integrins in lymphocyte intestinal homing, which eliminates pathogenic microorganisms and maintains intestinal immune tolerance via intestinal mucosal immunity. Research has shown that the recruitment of Shp1 by CD22, a lectin of B cells, inhibited β7 integrin endocytosis to promote the binding of β7 integrins with mucosal addressin cell adhesion molecule-1 (MAdCAM-1), which is expressed on postcapillary high endothelial venules in Peyer’s patches and mesenteric lymph nodes, thus initiating the intestinal homing of B cells [90,91]. Moreover, Ballet et al. highlighted that β7 integrins played a significant role in homing lymphocytes to gut-associated lymphoid tissue (GALT) via the CD22-Shp1 phosphatase axis, which ultimately facilitated the movement of B cells to the lamina propria of GALT [90].

Numerous studies have shown that integrins mediate the immune response of B cells, such as the synthesis of antibodies and antigen presentation. Through the function of the actin-binding protein vinculin, integrins on the B-cell surface maintain the stabilization of immune synapses, a transient structure centered on the TCR-MHC-antigenic peptide, to favor antigen presentation between B cells and T cells [7]. Additionally, Waffarn et al. demonstrated that the activation of αM integrins by type I interferon directly stimulated the adhesion and rapid redistribution of B lymphocytes to regional mediastinal lymph nodes, and this effect mediated the synthesis and accumulation of protective IgM at the site of infection [92]. Furthermore, the distribution of α4β1 integrins in the plasma membrane is induced by CD37, which is found in the germinal center (GC) of the spleen [93]. The gathered α4β1 integrins bind to vascular cell adhesion molecule 1 (VCAM-1) to activate the phosphatidylinositol-3-kinase (PI3K) signaling pathway and then phosphorylate the lipids on the plasma membrane to form the second messenger phosphatidylinositol (3,4,5)-triphosphate (PIP3), which activates the AKT signaling pathway [94]. The AKT signaling pathway ultimately mediates cell survival, thereby inhibiting apoptosis and increasing the production of plasma cells [93,94]. Likewise, toll-like receptor 9 (TLR9) also triggers the interaction of α4β1 integrins with VCAM-1, which promotes the retention and homing of circulating B cells in secondary lymphoid organs, and homing of the post-germinal central memory B cells into bone marrow or mucosal tissues to differentiate into plasma cells [95]. Research showed that IgA was secreted into the enteric cavity directly by the docking between αEβ7+ plasma cells and E-cadherin-expressing enterocytes, which identified αEβ7 integrins as an important target for the treatment of inflammatory bowel disease and provides new approaches for drug research and development, such as with natalizumab and vedolizumab. However, integrins also inhibit B-cell-mediated immune processes. It has been demonstrated that the deficiency of αV integrins promoted TLR signaling in GC B cells and enhanced the response to TLR ligand-containing antigens and influenza viruses, which led to affinity maturation of GC B cells and facilitated the generation of memory and plasma B cells, resulting in strong, high-affinity, and long-lived antibody responses [96].

Several autoimmune diseases have close relationships with integrins. As the most common central nervous system (CNS) demyelinating disease, MS is stimulated by the strong recruitment activity of β1 integrins. High levels of β1 integrins facilitate the accumulation of B cells in the central nervous system and the recruitment of Th17 and macrophages. These cells attack the myelin essential proteins and cause the destruction of the insulating sheath of nerve tissues, thereby leading to the onset of MS [8]. Nonetheless, some integrins also inhibit the occurrence of autoimmune diseases. In systemic lupus erythematosus (SLE), binding of αM integrins with CD22 downregulated B-cell receptor signaling to maintain autoreactive B-cell tolerance and further inhibited SLE progression [97]. Additionally, αV integrins prevented the development of SLE by inhibiting the interaction of TLRs on B cells with self-DNA and RNA [98]. Moreover, in non-autoimmune diseases, the fibronectin on bone marrow stromal cells in non-Hodgkin’s lymphoma patients adhered to α4β1 integrins on B cells, which can protect B cells from rituximab-induced apoptosis and conferred drug resistance [99]. Integrins in normal lymphocytes can also increase the susceptibility to viral infections. In Epstein-Barr virus (EBV) infection, the EBV adheres to nasal mucosa-associated lymphoid tissue memory B cells through the binding of glycoprotein BMRF-2 with α5β1 integrins, which inhibits the immunological function of B cells [100]. 

Integrins participate in regulating the activities of B cells in humoral immunity, such as adhesion, migration, and proliferation. Furthermore, they also mediate the immune response of B cells, including the synthesis of antibodies and antigen presentation. 

### 2.2. T Cells

Many significant activities of T cells, as a pivotal cell type in humoral and cellular immunity, are induced by integrins. Recent studies have shown that integrins mediate T-cell production and persistence [101]. It has been demonstrated that α4β1 integrins within tetraspanin-enriched microdomains shifted into a high-affinity state and accumulated through regulation of both CD9 and CD151, which promoted the activation of T cells [102]. Bromley et al. indicated that α1 integrins promoted the formation and persistence of CD8+ tissue-resident memory T cells and their abilities to secrete cytokines [101]. Moreover, the interaction of α5β1 integrins with CD154, which is an important marker of CD4+ T-cell activation, inhibited Fas/FasL signaling pathway-mediated CD4+ T and CD8+ T-cell death [103,104], and it decreased the production of caspase-8, all of which indicates that α5β1 integrins play an important role in T-cell survival and persistence in inflammatory diseases [103,104].

Research has elucidated that integrins can induce the movement of T cells, including their adhesion, migration, and accumulation at different sites. It has been reported that the flow of actin within T cells is regulated by the interaction of TCRs, integrins, and extrinsic signals [105]. T-cell activation mediated by TCR and ligand binding promotes actin centripetal movement, initiating T-cell movement [105]. This centripetal movement promoted integrin-ligand binding (αLβ2 integrins to ICAM-1 and α4β1 integrins to VCAM-1), thus reducing tyrosine in the TCR signaling pathway, which slowed actin centripetal flow and cell movement [105]. Numerous studies demonstrated that αLβ2 integrins facilitated the adhesion and movement of different T-cell types. Adel et al. showed that αLβ2 integrins were regulated by Galectin-9 to support naive CD4+ and CD8+ T-cell migration across endothelial cells in a glycan-dependent manner [106]. Similarly, research indicated that αLβ2 integrins regulated naive T-cell homing and controlled the inflammatory response through the Janus kinases 1 and 2 signaling pathways [107]. The binding of αLβ2 integrins with ICAM-1 induced the phosphorylation of protein tyrosine kinases (Lck) and zeta chain associated protein of 70 kDa (ZAP-70), which further promoted the conversion of intermediate-affinity αLβ2 integrins to high-affinity αLβ2 integrins, thereby increasing their binding with ICAM-1 [108,109]. Through the regulation of high-affinity αLβ2 integrins, the αLβ2/Lck/ZAP-70 complex initiated rapid adhesion and migration of T lymphocytes after encountering infection or injury stimuli [108,109]. In addition, β1 integrins are essential for memory CD4+ T-cell retention at different sites [110]. DeNucci et al. indicated that α1β1 integrins promoted the retention of CD4+ T cells in the bone marrow, lung, and epidermis [110]. Similarly, the synthesis of β1 and β7 integrins was increased by the stimulation of the proinflammatory cytokine HMGB1, which promotes the migration and adhesion of CD4+ T cells to endothelial cells [10]. Integrins facilitate CD4+ T-cell migration away from the lymph node subcapsular sinus and homing to lymph nodes [111]. Additionally, some regulating factors, such as kindlin-3 and all-trans-retinoic acid (ATRA), can activate integrins or upregulate the expression of integrins. The interaction between talin-1 and kindlin-3 and the tension generated by actin polymerization induced and stabilized the activities of β2 integrins and mediated the binding of integrins to ICAM-1, thereby facilitating the activation and migration of T cells [112,113]. Moreover, ATRA induced the upregulation of α4β7 integrins and the mucosal-homing receptor CCR9 by binding to retinoic acid receptors, which promoted T-cell migration and homing to the mucosal surface [114]. Furthermore, integrins also play an important role in regulating T-cell immunological function. Studies showed that the binding of αV integrins and αLβ2 integrins to ICAM-1 can favor antigen presentation [5,115]. αEβ7 integrins expressed on T cells regulate the migration of intestinal stem cells to interact with different types of immune cells and maintain intestinal homeostasis by interacting with E-cadherin [116].

The effect of integrins on T-cell immune activities determines its association with the occurrence and development of certain diseases. It was confirmed that β1 integrins caused the entry of Th17 cells into the CNS, which contributes to MS [117]. TNF-α and IL-1α induced the expression of melanoma cell adhesion molecules (MCAM), and the MCAM/CD146 signaling pathway triggered the phosphorylation of PLCg1, which led to the activation of β1 integrins on memory T cells and enhanced the infiltration of Th17 cells into the brain [117]. Similarly, it has been reported that α4 integrins promoted the entry of Th1 cells into the CNS to cause MS [118]. Additionally, Kanayama et al. indicated that the binding of tenascin-C and osteopontin with α9 integrins on dendritic cells (DCs) and macrophages in cooperation with TLR signaling promoted the production of IL-6 and IL-23, leading to the differentiation of naive CD4+ cells into Th17 cells and the development of arthritis [9]. Additionally, Bachsais et al. showed that α5β1 integrins on T-cell leukemia-lymphoma cells bound to CD154 and inhibited apoptotic events in leukemia cell lines through the activation of pro-survival pathways, including the p38, ERK1/2, and PI3K/Akt signaling pathways [119]. However, certain types of integrins prevent damage caused by immune cells or immunocompetent substances by promoting the generation, maintenance, and function of Tregs or by inhibiting the function of effector T cells. It has been demonstrated that integrins were linked to the cytoskeleton upon activation by talin and further recruited phosphatidylinositol phosphate kinase to maintain Treg homeostasis and peripheral tolerance, preventing autoimmunity [120]. Braun et al. indicated that the lack of αE integrins reduced the expression of FoxP3 in FoxP3+ Treg cells, which increased and recruited hapten-primed effector CD8+ T cells to sites of skin inflammation and draining lymph nodes, and enhanced the contact hypersensitivity reaction [121]. In addition, the expression of αV integrins on Tregs mediates TGF-β activation and inhibits the autoimmune response of effector T cells, promoting the accumulation of Tregs in the inflammatory gut, which plays an important role in inhibiting the development of colitis [122,123] (Figure 2).

Integrins play an important role in T-cell production and movement, including adhesion, migration, and gathering in various regions. At the same time, they are closely related to certain diseases, such as MS, arthritis, and T-cell leukemia.

### 2.3. Dendritic Cells

It is widely acknowledged that DCs are the most functional specialized antigen-presenting cells (APC) in the human body, and integrins play a significant role in their functional implementation, including migration, adhesion, maturation, and immunoregulation [124,125,126,127,128,129,130]. Previous studies showed that β2 integrins affected the podosome formation of DCs through their Ser 745 and Ser 756 amino acid residues [131], which correlated with actin-linkage of the GM-CSF receptor on DCs in the presence of kindlin-3 [130], and further influenced the migration of DCs. Similarly, Sie et al. indicated that α4 integrins differentially promoted the accumulation of DCs in the CNS and gut but had no effects on other parts of the body [127]. In addition, it has been reported that deficiency of kindlin-3 disrupted F-actin polymerization in DCs and inhibited MRTF-A nuclear shuttling, which prevented MRTF-A from co-activating genes with SRF in the nucleus, thereby affecting the adherent DC phenotype and integrin-mediated traction production [132]. Moreover, studies have confirmed that FAK and its downstream signaling molecules promoted β1 integrin-mediated production of mature ALDH1A2^high^ DCs, which indicates the role of β1 integrins in the differentiation and maturation of DCs activated by retinoic acid (RA) [133].

Numerous studies highlighted that integrins make substantial contributions to the immunological function of DCs, specifically in antigen uptake, antigen presentation, immunoregulation, and immunotolerance induction, which plays a critical role in connecting innate immunity with adaptive immunity. Researchers have indicated that αXβ2 integrins mediated phagocytosis of DCs activated by CCL21 in lymphatic endothelium by binding with the complement iC3b, which caused endocytosis and presentation of antigens [134]. Furthermore, it has been demonstrated that apoptotic cells promoted the expression of αV integrins on myeloid DC precursors, which activated TGF-β1 and further promoted CD103 expression and immunomodulatory functions [124]. Similarly, Duhan et al. have shown that αE integrins blocked virus-induced production of IFN-1 in conventional dendritic cells by inhibiting AKT activation and the mTOR signaling pathway to perform immunoregulatory functions [128]. 

Additionally, the involvement of integrins in various autoimmune diseases induced by abnormalities of DCs has been confirmed [126,133,135,136,137,138]. β1 integrins have been reported to promote the production of RA enzyme ALDH1A2 in GM-CSF-induced DCs through linkage with TLRs to effectively alleviate colitis [133]. Moreover, αVβ8 integrins on DCs initiated a CD4+ T-cell type 2 immune response activated by a high level of TGF-β and RA factors in the intestinal microenvironment and the TLR signaling pathway [125,139], thereby exposing mice to chronic intestinal parasite infection [135]. αVβ8 integrins also promoted IL-1β-induced and allergen-induced airway remodeling and secretion of CCL2 and CCL20 in adult fibroblasts through activation of TGF-β, leading to chronic obstructive pulmonary disease [136]. During the pathogenesis of inflammation, integrins function differentially [137,138,140]. In neuroinflammation, α4β1 integrins promoted the accumulation of plasmacytoid DCs into the CNS to relieve symptoms [126]. In Rotavirus infection, intestinal ΒATF3-dependent cDCs were shown to express α5β8 integrins, which enhanced IgA response by activating TGF-β [138]. Based on the importance of integrins to DCs, integrins are a promising target in immunotherapy.

Integrins play a significant role in the functional activities of DCs, including migration, adhesion, maturation, and immunoregulation. Furthermore, integrins also make great contributions to the immunological function of DCs, specifically in antigen uptake, antigen presentation, immunoregulation, and immunotolerance induction.

### 2.4. Natural Killer Cells

Numerous studies have shown that natural killer (NK) cells are a type of lymphocyte that can kill target cells without relying on antigens and antibodies and have an important role in the body’s innate immunity. It has been reported that integrins have significant effects on NK cell functions, including synthesis, proliferation, secretion, migration, natural killing, and immunosurveillance. Research has indicated that β2 integrins induced the phosphorylation of T-cell receptor ζ (TCRζ) and spleen tyrosine kinase (Syk) by binding to ICAM-1, which ultimately led to phospholipase C phosphorylation and paxillin-dependent perforin polarization [141]. Additionally, it has been shown that when ICAM-1 is bound to β2 integrins, talin is redistributed to the β2 integrin junction site and initiates two signaling pathways, which promotes the adhesion and migration of NK cells [142,143]. By binding to ICAM-1, β2 integrins induce the recruitment of talin and the actin nucleating protein complex Arp2/3, the result of which was a local increase in PIP2 levels. Furthermore, increased PIP2 recruited WASP to the β2 integrin junction sites, and WASP promoted Arp2/3-mediated actin polymerization, which led to NK cell adhesion to target cells and generated activation signals that resulted in the polarization of the actin cytoskeleton [144]. Intriguingly, it has been demonstrated that the tetraspanin CD53 regulated the response of activated NK cell receptors while promoting the activation of β2 integrins, which inhibited the effector function of NK cells [145]. 

Notably, integrins play a vital role in the immunoregulatory processes of NK cells, such as natural killing and immunotolerance [141,146,147]. Studies have shown that β1 integrins and NKp30 together acted as specific recognition receptors to activate the SFK-PI3K signaling pathway; at the same time, β1 integrins acted through the integrin-linked kinase (ILK)-Rac1 signaling pathway, thereby allowing NK cells to kill Cryptococcus neoformans [146]. Furthermore, it has been reported that αVβ3 integrins inhibited the inflammatory responses of NK cells, which are essential for the maintenance of decidua homeostasis and immunotolerance [148]. 

The expression of integrins on NK cells is intimately associated with pathological states. Numerous studies showed that α2β1 integrins contribute to the accumulation and proliferation of NK cells, indicating a crucial role in antiviral resistance [149]. In addition, during chronic inflammation, β1 integrins combined with the NKG2D and TRAIL pathways to mediate cytotoxicity against autoreactive CD4+ T cells and to exert immunomodulatory effects resulting from the inhibitory functions of the CD94/NKG2A pathway [147]. Furthermore, previous research showed that IL-2 stimulated the expression of α4β7 integrins on NK cells, which participated in the myocardial infarction repair process via promoting TNF-α-stimulated endothelial cell proliferation, enhancing collagen synthesis and angiogenesis, thereby reducing fibrosis within the infarcted myocardium, which may provide a novel approach for the treatment of myocardial infarction [150]. 

Integrins have significant effects on NK cells, including synthesis, proliferation, secretion, migration, natural killing, and immunosurveillance.

### 2.5. Neutrophils

Numerous studies indicate that integrins play a role in adhesion, aggregation, and movement. Lai Wen et al. reported that activated β2 integrins are involved in many immune functions, including the formation of inflammatory synapses at the neutrophil and endothelial interface, cell rolling, transport, and microparticle generation during transport [151]. It has been demonstrated that integrins drove the activation of PI3K and ARAP3, key regulators of integrin inactivation, promoting local integrin inactivation and turnover of substrate adhesion [152]. In addition, β2 integrins mediated neutrophil cross-cell migration induced by Rap1b deletion, PI3K-AKT activation, and the enhancement of invadopodia-like protrusions [153]. Bromberger et al. showed that β1 and β3 integrins on neutrophils were activated by talin-1, which was recruited to the plasma membrane through direct Rap1/talin-1 interaction [154]. The activation of β1 and β2 integrins was found to be facilitated by Radil, a novel Rap 1 effector, which rapidly translocates from the cytoplasm to the plasma membrane in Rap1a-GTP dependent manner [155]. Additionally, research indicated that β2 integrins bind to talin-1 and kindlin-3 to mediate the aggregation of neutrophils [156,157]. Skap2 was shown to regulate actin polymerization as well as the binding of talin-1 and kindlin-3 to the cytoplasmic domain of β2 integrins, thus promoting the activation of β2 integrins and neutrophil recruitment [156]. Integrin-linked kinase (ILK) promoted the membrane targeting of protein kinase C (PKC) and chemokine-induced kinase activity, thereby promoting the binding of kindlin-3 to the cytoplasmic domain of β2 integrins, which induced binding of extend (E+) headpiece (H-) β2 integrins to ICAM-1 expressed on activated endothelial cells in cis [157,158]. β2 integrins facilitate the arrest, polarization, and crawling of neutrophils on ICAM-1 and ICAM-2, which is a prerequisite for neutrophils to cross the inflamed blood-brain barrier [159]. The binding activity of αMβ2 integrins to their ligand is regulated by protein disulfide isomerase (PDI), which promotes the recruitment of neutrophils to the sites of inflammation during vascular inflammation [160]. Silva et al. reported that the involvement of αMβ2 integrins in neutrophil accumulation was mediated by the activation of fibrin [161]. Moreover, several studies highlighted that the interaction of β2 integrins with cytohesin-1 and galectin-9 promoted neutrophil capture in human blood and adhesion to endothelial cells [162,163]. Pruenster et al. demonstrated that β2 integrin-dependent extracellular regulator MRP8/14 induced slow rolling and adhesion of neutrophils during inflammation in vivo [164]. Previous studies elucidated that the expression of β2 integrins on the plasma membrane of neutrophils is regulated by CD37, which promotes the cytoskeletal function of integrin-mediated adhesion downstream [165]. Furthermore, the dynamic redistribution of β2 integrins and non-muscle myosin (IIA) was shown to be coordinated by chemotactic signals generated by the eicosanoid leukotriene B4 (LTB4) and its receptor BLT1 to promote the arrest and extravasation of neutrophils and migration to the sites of inflammation [166]. In addition, numerous studies indicated that β1 and β3 integrins activated by vitronectin suppressed the apoptosis of neutrophils [167]. Roberta et al. showed that the binding of α9β1 integrins to VLO5, a snake venom disintegrin, induced the degradation of the pro-apoptotic protein Bad and the expression of the anti-apoptotic protein Bcl-xL, effectively reducing the spontaneous apoptosis of neutrophils [168]. 

Numerous studies have elucidated that integrins regulate neutrophil-associated inflammatory responses and immune killing, which can be the etiology of some diseases. It was reported that αLβ2 integrins triggered the extravasation of neutrophils into areas of the CNS with amyloid-β (Aβ) deposition, thereby initiating Alzheimer’s disease [169]. Research showed that αM integrins regulated by Rho GTPase Cdc42 in the uropod of neutrophils controlled the polarity of neutrophils and mediated neutrophil migration to the inflamed lungs [170]. Similarly, Herbert et al. reported that β2 integrins bound to ICAM-1 on epithelial cells, which mediated transepithelial migration of neutrophils during respiratory syncytial virus infection, causing airway damage [171]. It was demonstrated that the integrin-related G-protein coupled receptor (GPCR) signaling pathway activated neutrophils and induced the formation of neutrophil extracellular traps in acute lung injury [172]. Additionally, β2 integrins promoted the recruitment of neutrophils to lung tissue to cause inflammatory lung disease, which was inhibited by Siglec-E [173,174].

Integrins make great contributions to the adhesion, aggregation, and movement of neutrophils. Moreover, they also participate in neutrophil-associated inflammatory responses and immune killing.

### 2.6. Macrophages

Numerous studies have elucidated that integrins play a critical role in macrophages, a type of leukocyte belonging to the mononuclear phagocytic system, including their activation, adhesion, migration, phagocytosis, proinflammatory, and anti-inflammatory effects [175]. Research has shown that αV integrins contribute to the development of dry eye disease (DED) by binding with vitronectin (VTN) and activating NF-kB, which induces inflammatory gene expression in the bone marrow-derived macrophages [176]. Moreover, previous studies indicated that integrins controlled the activation and differentiation of macrophages. For example, α2β1 integrins inhibited the induction of M2 macrophages and maintained the macrophage shape [177,178]. Experiments have elucidated that αDβ2 and αMβ2 integrins are regarded as important targets controlling the balance between M1 and M2 macrophages, which is critical in the development of chronic inflammation [179]. In addition, it was reported that α1β1 integrins mediated adhesive interactions between macrophages and inflammatory lesions by binding with collagen IV in the ECM [180]. For example, a lipotoxic insult stimulated the phosphorylation of p38 in hepatocytes, causing hepatocytes to release β1 integrins as extracellular vesicles, which mediated macrophage adhesion to liver sinusoidal endothelial cells by binding with VCAM-1, leading to hepatic inflammation [181]. Additionally, numerous studies have shown that integrins participate in the molecular mechanism of macrophage migration to inflammatory sites through outside-in signaling pathways. For example, it has been reported that β1 integrins interacted with hydrogen sulfide (H_2_S) and activated the Src/FAK/Pyk2/Rac signaling pathways in macrophages, thereby promoting macrophage migration into an infarct area [182]. Additionally, α4β1 integrins have been reported to mediate macrophage recruitment to atherosclerotic plaques by binding with VCAM-1 on vascular endothelial cells [183]. Similarly, insulin-like growth factor-1 (IGF-1) expressed in atheroma bound with αVβ3 integrins activated the PI3 kinase/PKC/p38 signaling pathway, an essential step in the progression of atherosclerosis [184]. Osteopontin (OPN), as a potent chemokine, has been confirmed to affect macrophages via two distinct pathways. OPN binds with αVβ5 integrins on macrophages, which is critical for maintaining the M2 macrophage gene signature and development of mesenchymal glioblastoma; conversely, OPN could interact with α4 integrins to mediate macrophage chemotaxis [185,186]. 

Previous experiments highlighted that integrins have both pro- and anti-inflammatory roles in the inflammatory responses related to diseases. It has been reported that αDβ2 integrins on M1 macrophages interacted with the ECM of peripheral tissues, thereby enhancing the accumulation of pro-inflammatory macrophages at atheromas [187,188]. In the molecular mechanism of cardiac inflammation and hypertrophy, β2 integrin transcription was enhanced by interacting with integrin β2 (ITGβ2), which is a complex composed of KDM3A, MRTF-A, and Sp1, and thereby promoted the inflammatory responses [189]. In addition, studies have indicated that bile acids secreted by damaged biliary epithelial cells (BECs) promoted the recruitment of CCL2 and the activation of DAMPs. CCL2 then bound to CCR2 on macrophages and induced inflammation, which promoted the activation of TGF-β1 and led to portal fibrosis; at the same time, ITGF-β6 and ITGF-β6 ligands were upregulated, resulting in the proliferation of BECs [190]. Additionally, it has been reported that integrins in macrophages combine with other immune cells to contribute to diseases. For example, the expression of β2 integrins on macrophages induces the Rac1/reactive oxygen species (ROS)/non-canonical NLRP3 inflammasome/Prp-IL-1β/IL-1β signaling pathway. IL-1β binding to IL-1R on the surface of Group 3 innate cells promoted the expression of IL-22, which participated in the development of C.rodentium-induced colitis [191]. In contrast, numerous studies have indicated that integrins enhanced the anti-inflammatory functions of M2 macrophages. For example, β1 integrins were shown to induce anti-inflammatory effects on lung alveolar epithelial cells by inhibiting ROS production and downregulating NF-kB-dependent chemokines, such as CCL2, which mediated the recruitment of macrophages to attenuate lung inflammation and emphysematous remodeling [192]. In addition, previous experiments showed that αVβ5 integrins are involved in M2 macrophage polarization by interacting with PPARγ stimulated by convallatoxin, which increased the level of IL-10 and decreased the level of pro-inflammatory factors, such as IL-6 and TGF-α, thereby attenuating atherosclerosis [193]. It is also worth mentioning that αVβ3 integrins were shown to mediate anti-inflammatory functions of M2 macrophages via the PI3 kinase/AKT-dependent NF-kB signaling pathway, which is critical for the prevention of and recovery from inflammatory states [194].

Integrins play a critical role in macrophage biology, including activation, adhesion, migration, phagocytosis, proinflammatory and anti-inflammation effects, which are related to several diseases, such as hypertension, atherosclerosis, and lung inflammation.

## 3. The Roles of Integrins in Cancer

### 3.1. Positive Modulation of Integrins in Cancer Development

#### 3.1.1. Evading Growth Suppressors

Integrins have been shown to play an important role in the overgrowth characteristics of cancer, which are mediated by a combination of genetic and environmental factors. Previous studies showed that αVβ3 integrins, as a receptor of thyroid hormone, mediated malignant T-cell angiogenesis by enhancing the expression and secretion of VEGFB and VEGFA, which in turn promoted endothelial cell migration and T-cell lymphoma proliferation [14]. In addition, it has been reported that α4β1 integrins on cancer cells promoted tumor growth by interacting with talin and paxillin, which are induced by the pro-inflammatory cytokine IL-1β and the chemokine stromal cell-derived growth factor 1 alpha (SDF-1α) [195]. Furthermore, β4 integrins promote the differentiation of cancer stem cells (CSCs) induced by differentiation antigen, which furthers cancer cell proliferation [196]. Notably, studies have elucidated that αVβ6 integrins were rapidly upregulated on epithelial cells during tissue injury and facilitated the differentiation of hepatic progenitor cells into bile duct cells and hepatocytes by activating TGF-β1, thereby promoting ductal fibrosis and, consequently, tumorigenesis [197,198]. Moreover, previous studies showed that αV integrins on M2-macrophages were activated via the TGF-β/smad2/3 signaling pathway, thereby mediating the stemness traits of pancreatic cancer cells [199]. Similarly, αVβ5 integrins on M2-macrophages secreted TGF-β and activated the Src/Stat3 signaling pathway, which promoted the maintenance of glioma stem cells (GSCs) and glioma growth [200] (Figure 3).

#### 3.1.2. Invasion and Metastasis

Integrins are essential to cancer metastatic characteristics, which is one of the major challenges in the treatment of cancer. Research has shown that β2 integrins increased the adhesion and invasiveness of B-cell acute lymphoblastic leukemia cells by promoting extramedullary exudation [16]. In addition, αV integrins have been reported to be overexpressed in gastrointestinal cancer cells, which promote cancer cell invasion and metastasis, resulting in a poor prognosis for patients [201]. In addition, research demonstrated that αV integrins reshape the tumor microenvironment by activating TGF-β, resulting in increased epithelial-mesenchymal transition, angiogenesis and invasion, cancer-associated fibroblast formation, and suppression of T-cell-mediated immune surveillance [202]. One study showed that cancer cells also changed the tissue-specific homing patterns and the characteristics of immune cells by secreting exosomes that were rich in specific integrins, which contributed to the formation of a microenvironment that induced tumor occurrence. This process induced several biological functions, such as increased vascular permeability, angiogenesis, and chronic inflammation, which cultivated a microenvironment that allowed subsequent cancer-homing retention and growth [203]. Other findings revealed that in B-cell chronic lymphocytic leukemia (B-CLL), JAK2 on the surface of B cells promoted cell adhesion and survival in bone marrow stromal niches by activating αLβ2 and α4β1 integrins, mediating B-cell-dependent adhesion and depression in secondary lymphoid-like organs [204]. 

#### 3.1.3. Immune Tolerance

The immune tolerance properties of cancer cells remain challenging in terms of clinical antitumor drug therapy, including immunological escape and drug efflux. Investigations elaborating the immune escape characteristics of cancer cells showed that αVβ3 integrins were upregulated on a variety of drug-resistant cancer cells, which induced immune tolerance and promoted immune escape via activating ATM/Chk2 and NF-kB-mediated pathways, thereby impairing the ability of DCs to cross-prime antigen-specific T lymphocytes [15]. Moreover, research demonstrated that the αVβ8 integrins on tumor cells promoted tumor-immune evasion via activating latent TGF-β on the surface of T cells, subsequently inducing the transformation of T_regs_ and thereby producing an immunosuppressive effect [93,205]. Moreover, it has been reported that αVβ6 integrins on CD4+CD25+ T cells also inhibited the cytotoxic effects of CD8+ T cells by activating TGF-β, which reduced pro-inflammatory tumor-associated macrophage aggregation towards tumors, thereby suppressing anti-tumor immunity [198,206,207]. Additionally, T_regs_ were shown to work in tandem with cancer cells to produce biologically active TGF-β and create an immunosuppressive microenvironment, the biological effect of which is tumor growth [208]. In the case of NK cells, as αV integrins on GSCs interacted with CD9 and CD103 on their surface, GSCs were induced to produce and release TGF-β due to the cell-to-cell contact, leading to impaired NK cell lysis and increased tumor infiltration. Additionally, several studies highlighted that α4 integrins activated by PI3Kγ promoted the polarization and aggregation of myeloid-derived suppressor cells (MDSCs) with immunosuppressive properties and tumor-associated macrophages in tumors during tumor progression, thereby inhibiting T-cell-mediated anti-tumor immune responses and promoting tumor progression [209]. Moreover, it has been reported that β1 integrins promoted adriamycin resistance in human T-cell acute lymphoblastic leukemia by activating Fak-related proline-rich tyrosine kinase 2 (PYK2), which in turn activated the ABCC1 drug transporter and facilitated adriamycin efflux [210]. 

Integrins positively modulate cancer development by supporting growth suppressor evasion, invasion, metastasis, and immune tolerance. 

### 3.2. Negative Modulation of Integrins on Cancer Development

Integrins can serve as immune-binding targets for APCs. Previous research showed that αVβ3 integrins are upregulated in tumor cells resistant to various drugs in which ATM/Chk2-and NFκB-mediated pathways are activated and act as an immune target for DCs [15]. Several studies have highlighted that integrins promote the recruitment and infiltration of CD8+ T cells and CD8+ tissue-resident memory T cells (TRMs) in tumor tissues, enabling T cells to release cytokines and cytotoxic particles, which induce the apoptosis of tumor cells [211,212]. Additionally, αLβ2 and α4β1 integrins contribute to CD8+ T-cell recruitment, adhesion, spreading, cytolytic activity, co-stimulation, and infiltration in tumor tissues [18,213]. Mahurin et al. indicated that αLβ2 integrins colocalized with and were activated by layilin (a c-type membrane glycoprotein containing the lectin domain) on CD8+ T cells in human melanoma, which enhanced cell adhesion and CD8+ T-cell accumulation in tumors, and inhibited tumor growth [214]. It has been confirmed that TGF-β produced by activated immune cells activated αEβ7 integrins on CD8+ T cells, which enhanced αEβ7-dependent T-cell adhesion and signal transduction to induce the recruitment of CD8+ T cells to epithelial tumor islets [215]. TGF-β activates through αEβ7 integrins by binding between TGF-β and its receptors (TGFBR), which induces the recruitment and phosphorylation of ILK; ILK then interacts with the intracellular domain of αEβ7 integrins, leading to protein kinase B (PKB) activation, thus initiating integrin-based inside-out and outside-in signaling pathways [215]. Similarly, αEβ7 integrins facilitate the retention of CD3+ CD8+ TRMs, the cytotoxic immune synapses of which bind to specific cancer cells in epithelial tumor islands, leading to TCR-dependent target cell killing [211,212]. In addition, integrins can promote the interaction between TCRs of CD8+ T cells and tumor cells to enhance the function of cytotoxic T-lymphocytes. It has been demonstrated that αLβ2 integrins on tumor cells increase the binding power between cytotoxic T cells and tumor cells by binding to TCRs, thus promoting cytotoxic T cells to exert anti-tumor immune effects [216] (Figure 4). 

Integrins make great contributions to the negative modulation of cancer development. They can not serve as immune-binding targets for APCs, but they can enhance T-cell recruitment in tumors, which inhibits tumor growth.

Moreover, numerous studies demonstrated that integrins can inhibit tumor growth by promoting the cytotoxic effects of NK cells. Research has elucidated that αLβ2 integrins and actins located in NK cells are activated by the expression of 2B4 receptors on NK cells, mediating their rapid adhesion to tumor cells and participation in the early stage of cytotoxicity [17]. Anikeeva et al. reported that αV integrins on tumor cells promoted antibody-dependent cell-mediated cytotoxicity in NK cells in the absence of pro-inflammatory cytokines, resulting in the production of particles and the dissolution of cancer cells [217]. Integrins also promote the recruitment and phagocytosis of macrophages. It has been reported that inflammation promoted the expression of αL and αX integrins in macrophages, thereby inhibiting CD47 overexpression in tumor cells, which prevented checkpoint SIRPa-CD47 from exerting anti-phagocytic effects [218]. In addition, previous studies showed that β2 integrins on macrophages interacted with signaling lymphocytic activation molecule-7 (SLAM7) and activated immunoreceptor tyrosine-based activation motifs, which initiated actin polarization and induced phagocytosis [218]. Moreover, it has been reported that β4 integrins promoted the activation of TGF-β1 and induced the upregulation of laminin-5 and Tie2, which led to enhanced macrophage adhesion [219].

### 3.3. Integrin-Targeted Immunotherapy 

#### 3.3.1. Chimeric Antigen Receptor T-Cell Immunotherapy

CAR T-cell immunotherapy relies on the combination of tumor CAR and patient-derived T cells to enhance the targeting specificity of T cells, which is thus widely applied for the treatment of hematological tumors, such as B-cell lymphoblastic leukemia and lymphoma [220,221]. αvβ6-Binding peptide (A20) fusion with a second-generation CD28/CD3ζ signaling domain or with a fourth-generation CD28/4-1BB/CD27/CD3ζ signaling domain to form A20-2G CAR and A20-4G CAR, respectively, achieves highly selective targeting of αvβ6 [222,223]. Moreover, Whilding et al. demonstrated that the addition of cytokine IL-4 in vitro amplified CAR T cells, which then targeted the tumor-associated αVβ6 integrins to improve T-cell efficacy and availability [223]. Similarly, the activated conformation of β7 integrins has been demonstrated to be a new and specific target of CAR T cells for the treatment of multiple myeloma (MM) [224]. Additionally, CAR T-cell therapy has achieved remarkable results in the treatment of hematological malignancies [221]. The unique specificity of CAR T cells allows them to accurately destroy cancer cells containing corresponding tumor-associated antigens, antigens without damage to normal tissues. Moreover, the augmentation of the immune surveillance function of CAR T cells mediates the recognition of cell-surface molecules without the aid of HLA expression [221]. However, numerous challenges still exist with CAR T-cell immunotherapy, limiting the therapeutic efficacy of CAR T cells in solid tumors, including severe life-threatening toxicities, modest anti-tumor activity, antigen escape, restricted trafficking, and limited tumor infiltration. In order to overcome these significant challenges, innovative approaches will need to be implemented to produce more powerful CAR T cells with anti-tumor activity and decreased toxicity in the near future [225].

#### 3.3.2. Oncolytic Virotherapy

Oncolytic virotherapy is conducted by modifying the specific virus to mediate immune killing or to disrupt the tumor blood supply in the tumor microenvironment (TME) [226,227]. It has been demonstrated that Ad5NULL-A20, a target-modified αVβ6 integrin-selective oncolytic adenovirus, inserted an A20 peptide into its capsid protein to target αVβ6 integrins for the treatment of pancreatic and breast cancer [228,229]. Additionally, Pesonen et al. reported that the RGD-4C-modified oncolytic adenovirus Ad5D24-RGD entered tumor cells through αV integrins highly expressed in advanced tumors [230]. Ad5D24-RGD facilitated immune-killing by incorporating granulocyte-macrophage colony-stimulating factor (GM-CSF), a powerful immune activator [230]. Other research indicated that the oncolytic herpes simplex virus-1 (oHSV) produces OS2966, a therapeutic antibody that blocks the migration of macrophages by targeting the β1 integrins receptor on macrophages [227]. Compared with many other treatments, oncolytic virotherapy produces fewer adverse effects [231] (Table 2). However, tumor cells make use of the body’s self-protection mechanism to block the specific virus from entering and replicating in cells, thereby attenuating the therapeutic effects of oncolytic virotherapy. To overcome these limitations, having a deeper understanding of the mechanism by which tumor cells protect themselves from oncolytic viruses is necessary. 

#### 3.3.3. Immune Checkpoint Inhibitor Therapy

ICI therapy refers to the reduction of T-cell immune suppression by inhibiting the immune checkpoints on the surface of T cells and enhancing the function of T-cell immune surveillance and tumor killing [202,236]. Several studies have confirmed that downregulation of αVβ3 integrins can inhibit the expression of PD-L1 mediated by the BRAF/TAK1/ERK/ETV4 signaling pathway, thus inhibiting the immune escape of cancer cells [235,236,242]. Other research showed that the inhibition of αVβ3 integrins downregulated the production of active TGF-β, thus promoting the invasion and infiltration of local cytotoxic T cells and tumor cell killing [202]. Similarly, Busenhart et al. showed that the inhibition of αVβ6 integrins stimulated T-cell antitumor responses and enhanced immune checkpoint blockade therapy in colorectal cancers [207]. Currently, ICI has a widely applied as a first and second-line treatment of advanced hepatocellular carcinoma. Nevertheless, there are still several limitations that hinder the therapeutic potency of ICI therapy, including the immuno-oncology (IO) bubble phenomenon, drug hyperinfiltration cycle, peculiar side effects, and paradoxical responses. Based on these challenges, many attempts have been made to improve this cancer treatment in the clinic. Changing to more rational, biology-based trials that study meaningful biomarkers tied to IO-specific outcomes might improve the clinical outcomes. Moreover, it may also be helpful to integrate modeling and simulation strategies into trial designs, which would be a prerequisite for improving therapeutic efficacy and survival [243].

#### 3.3.4. Other Strategies

In addition to the three therapies mentioned above, there are other cancer treatments available. It has been indicated that CSCs can be used as immunotherapy targets [196,244]. The expression of β4 integrins on CSCs promotes tumor growth and metastasis, and CSCs also influence the maturation of DCs [196,244]. In addition, inhibition of JAK2 or BTK, downstream effectors of CXCL12, inhibits integrin activation and suppresses tumor cell adhesion, survival, and spread in B-cell chronic lymphocytic leukemia (B-CLL) [204]. Furthermore, Nadine et al. demonstrated that NKG2A blockers prevented the ligand Qa-1/HLA-E on tumor cells from binding to the inhibitory receptor NKG2A on the surface of CD8 + T cells, avoiding adaptive immune tolerance development [245]. Similarly, OPN blockers prevent plasma OPN from interacting with tumor-derived integrins to inhibit tumor growth, invasion, and metastasis [246]. Nanotherapy is a therapeutic approach in which nanoparticles are loaded with antineoplastic drugs and delivered to tumors. αVβ3 integrin-specific cyclic arginine–glycine–aspartate nanoparticles actively bind to neutrophils in the blood before migrating to the tumor, hitchhiking to the tumor location to deliver drugs to the tumor islet [237]. Remarkably, natalizumab is widely used to treat MS; however, when present in the body for a long time, natalizumab affects the degranulation of the melanoma cells by α4β1 integrins, thereby promoting the development of melanoma [206]. Moreover, strategies involving the use of integrin-simulating targeting molecules (in particular, targeting peptides) are used to develop drug delivery systems targeting cancer cells. For example, the well-known VHPKQHR peptide was identified by Kelly et al. through a phage display technique and is homologous to α4β1 integrins [247,248,249,250].

Integrin-targeted immunotherapies are shown to be suitable therapeutic approaches for cancer treatment, including CAR T-cell immunotherapy, oncolytic virotherapy, and ICI therapy.

## 4. Discussion

Integrins, as universal transmembrane heterodimeric proteins, play a crucial role in both intracellular and extracellular signaling pathways [8]. Numerous studies indicated that integrins have an important role in the regulation of immune cell structures and functions, including migration, adhesion, proliferation, immunotolerance, and immunosurveillance. At the same time, abnormalities of immune cell functions can lead to cancer, the mechanisms of which are twofold. On the one hand, integrins are involved in the regulation of cancer cell proliferation, invasion, migration, and angiogenesis; on the other hand, integrins promote immune cell aggregation to mediate the elimination of tumors. These findings on the underlying mechanisms of integrins in immune cell functions and cancer progression will provide new ideas for combined immunotherapy for cancer patients in the near future. 

Based on the effects of integrins on immune cells and cancer progression discussed in this review, a wide range of prospects can be investigated to discover suitable therapeutic methods. CITs, currently a hotspot of cancer treatment nowadays research, have been studied extensively, such as in CAR T-cell immunotherapy, oncolytic virotherapy, and ICI therapy. Compared to traditional cancer treatments, CITs show the advantages of strong effects, good tolerance, and lower toxicity. Although much progress has been made in the field of CIT, there are still numerous challenges to achieving a good prognosis due to the wide variety of integrins, their poor specificity, the varying activity of immune cells, and the nature of cancer cells themselves [251]. For example, the activation of anti-tumor immunity in CAR T cells results in a strong driving effect of cytokines, potentially culminating in macrophage activation syndrome and hemophagocytic lymphohistiocytosis [221]. Similarly, neurotoxicity and endothelial dysfunction can be caused by CAR T-cell therapy [221]. In addition, great challenges have been faced in the field of solid tumor treatment due to the insufficient and atypical variety of molecular targets of the TME in which CAR T cells act and due to the existence of immunosuppressive cytokines in the TME [221,231]. Furthermore, research showed that the number of circulating activated T cells targeting autoantigens can be increased by ICI therapy, which may lead to autoimmunity [252]. This autoinflammatory toxicity contributes to diseases of the CNS and peripheral nervous system, such as encephalitis, MS, myasthenia gravis, and peripheral neuropathy [252].

Although CIT is not clinically ideal, it can be improved in several ways. First, experimental models must be improved because current models may not accurately represent clinical effects. In preclinical models, both animal models and tumor transplants cultured from cell lines cannot replace and mimic human tumors in their biological function. Moreover, preclinical trials do not fully simulate conditions in which an integrin inhibitor is used as the main therapeutic approach to eliminate most tumor tissues or as adjuvant therapy for the reduction of residual diseases and circulating cancer cells. Second, a greater understanding of how molecular and cellular drivers function during primary versus secondary immune escape would make CIT more effective. For example, the effects of integrin expression at different stages of various cancers are not well understood. Third, personalized approaches can be maximized through composite biomarkers. The advantages of integrin inhibitors, such as their low side effects, have led to their rapid development as a popular treatment for cancer today, and their combination with conventional drugs could be further investigated. Moreover, the gene therapies that target various integrins are potentially highly effective cancer treatments. Advances in technology and treatment methods have made it possible to gain a better understanding of immune cells and cancer. Moreover, these initiatives provide further opportunities to promote more effective and less costly care for cancer patients. Consequently, we conclude that integrins have a great impact on immunity and cancer development, providing new clues into clinical CITs.

## Figures and Tables

**Figure 1 ijms-24-06170-f001:**
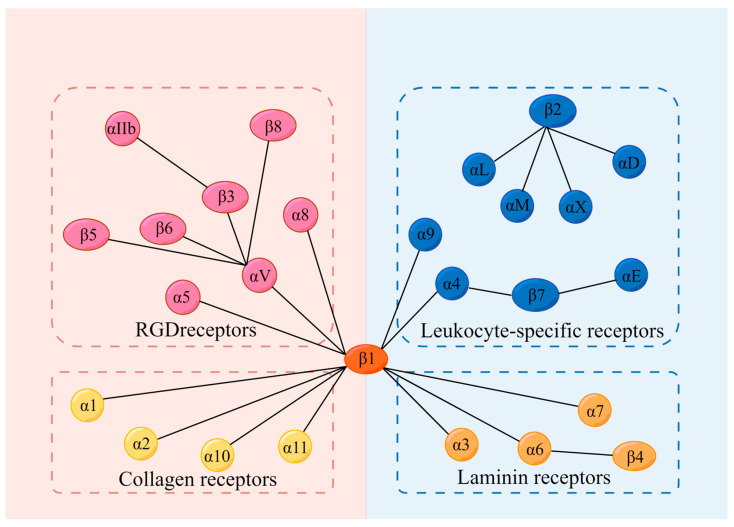
Integrins and their 4 categories of ligands; 18 α subunits and 8 β subunits constitute 24 integrins in humans, which can be divided into 4 categories according to their receptors. They are RGD receptors, leucocyte-specific receptors, collagen receptors, and laminin receptors. RGD: Arg-Gly-Asp.

**Figure 2 ijms-24-06170-f002:**
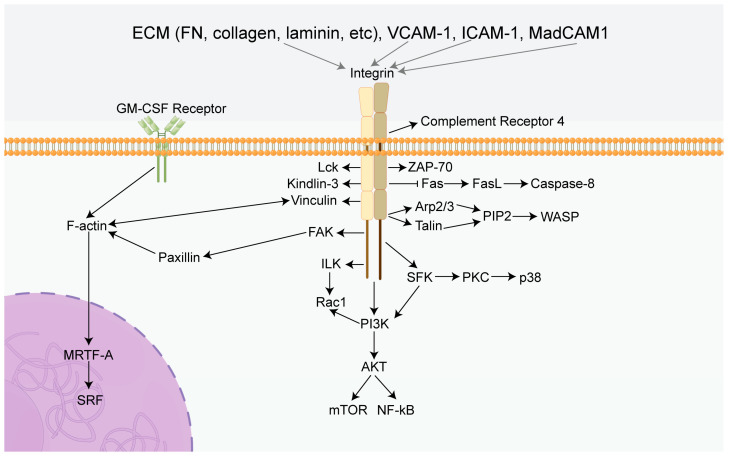
Summary diagram of the effects of integrins in immune cells. Integrins can mediate several classical signaling pathways, including PI3K/AKT/mTOR, ILK/Rac1, and Fas/FasL. Activation of these specific pathways leads to a cascade reaction inducing the characteristics of immune cells, including proliferation, survival, migration, adhesion, and immunotolerance. FN: Fibronectin, GM-CSF Receptor: Granulocyte-macrophage colony-stimulating factor receptor, FAK: Focal adhesion kinase, PI3K: Phosphatidylinositol 3-phosphokinase, AKT: Protein kinase B, mTOR: Mammalian target of rapamycin, ILK: Integrin-linked kinase, Rac1: Rac Family Small GTPase, Fas: TNF receptor superfamily, FasL: Ligand of TNF receptor superfamily, PIP2: phosphatidylinositol (1,2,3) -triphosphate, Arp2/3: Actin nucleating protein complex, WASP: Wiscott aldridge syndrome protein, MRTF-A: Myocardin-related transcription factor-A, SRF: Serum response factor, SFK: SRC-Family protein tyrosine kinase, VCAM-1: Vascular cell adhesion molecule 1, ICAM-1: intercellular adhesion molecule 1, MadCAM1: Mucosal address in cell-adhesion molecule-1, Lck: Protein tyrosine kinase, ZAP-70: Zeta chain associated protein of 70 kDa, PKC: Protein kinase C, NF-kB: Nuclear factor-k-gene binding.

**Figure 3 ijms-24-06170-f003:**
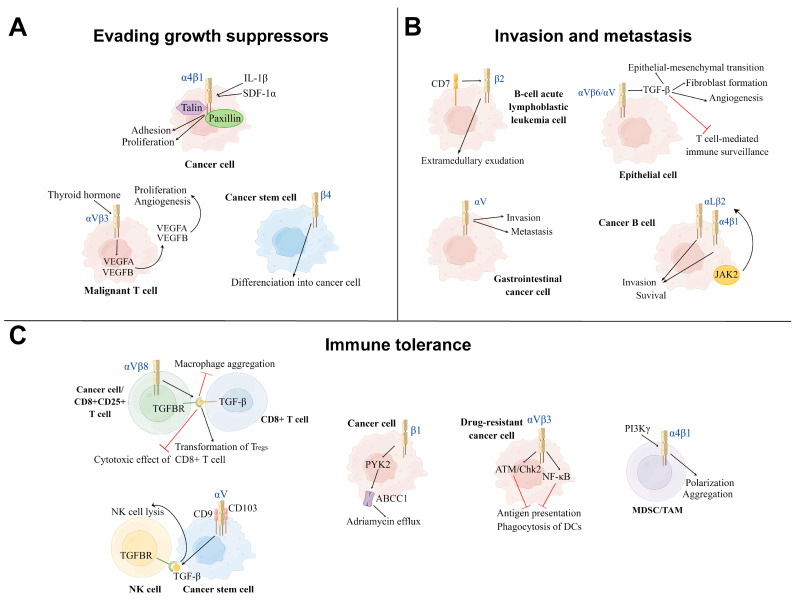
Three main mechanisms of positive modulation of integrins on cancer development. (**A**): Evading growth suppressor: Integrins can achieve evading growth suppressor by promoting cancer cell proliferation, differentiation, aggregation, and angiogenesis in the tumor microenvironment (TME). Under the influence of thyroid hormone, αVβ3 Integrins promote the proliferation and angiogenesis of the malignant T cell. The cytokines IL-1β and SDF-1α induce α4β1 integrins to interact with talin and paxillin to facilitate tumor growth. β4 integrins promote the differentiation of cancer stem cells into tumor cells. (**B**): Invasion and metastasis: Integrins mediate the invasion and metastasis of different types of cancer cells to other sites, leading to disease deterioration. β2 integrins and αV integrins promote, respectively, the invasion and metastasis of B-cell acute lymphoblastic leukemia cells and gastrointestinal cancer cells. αV integrins on the surface of epithelial cells can reshape the TME via activating TGF-β. The cancer B-cell in B-cell chronic lymphocytic leukemia (B-CLL) increases its survival and invasion by the activation of αLβ2 and α4β1 integrins via the JAK2 pathway. (**C**): Immune tolerance: Part of integrins on cancer cells inhibit the function of immune cells from achieving immune tolerance. αVβ3 integrins on the drug-resistant cancer cells inhibit the function of DCs via ATM/Chk2 and NFKB-mediated pathways. αVβ3 and αV integrins induce the process of TGF-β and TGFBR binding to suppress the immune killing effect of CD8+ T cell and NK cell, respectively. α4 integrins activated by PI3Kγ induce the polarization and aggregation of myeloid-derived suppressor cells (MDSCs) and TAMs to inhibit T-cell-mediated anti-tumor immune responses. In human T-cell acute lymphoblastic leukemia (ALL), β1 integrins activate the ABCC1 drug transporter of adriamycin via the PYK2 pathway. TH: Thyroid hormone, SDF-1α: Stromal cell-derived growth factor 1 alpha, DCs: Dendritic cells, NK cell: Natural killer cell, TGFBR: TGF-β receptor, MDSC: myeloid-derived suppressor cell, TAM: Tumor-associated macrophage, PYK2: Proline-rich tyrosine kinase 2.

**Figure 4 ijms-24-06170-f004:**
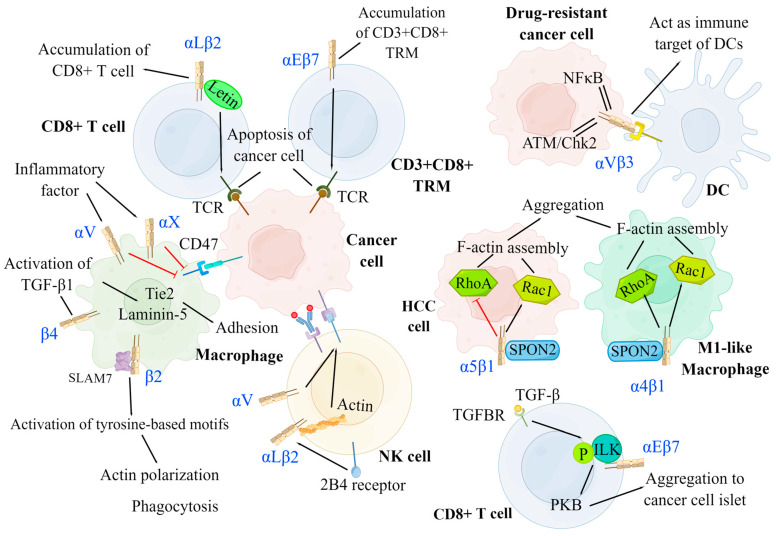
Negative modulation of integrins on cancer development. Integrins can serve as an immune target for DCs to promote antigen presentation. By interacting with several signaling molecules, integrins facilitate the migration, aggregation, and killing of immune cells, including DCs, T cells, NK cells, and macrophages. DCs: αVβ3 integrins on drug-resistant cancer cells have increased expression for ATM/Chk2-and NF-κB-mediated pathways and act as the immune target of DCs. T cell: αLβ2 integrins on CD8+ T cell and αEβ7 integrins on CD3+CD8+ TRM separately induce the accumulation of CD8+ T cell and CD3+CD8+ TRM at the cancer region and promote the binding between TCR and cancer cell to result in cancer cell apoptosis. αEβ7 integrins on CD8+ T cells are activated by TGF-β via phosphorylated ILK and facilitate the aggregation of CD8+ T cells to cancer islets. NK cell: αLβ2 and αV integrins can enhance the cytotoxicity of NK cells by promoting the adhesion between NK cells and tumor cells and promoting ADCC, respectively. Macrophage: The expression of αL and αX integrins are promoted by inflammatory factors, which inhibit the amount of CD47 on cancer cells, thus preventing the function of checkpoint SIRPa-CD47. The signal of SPON2-α4β1 on M1-like macrophage activates both RhoA and Rac1 to facilitate its aggregation. Conversely, the signal of SPON2-α5β1 on HCC cells inhibits the aggregation of HCC cells by suppressing RhoA. CD3+CD8+ TRM: CD3+CD8+ tissue-resident memory T cells, NK cell: Natural killer cell, DC: Dendritic cell, TGFBR: TGF-β receptor, PKB: Protein kinase B, ILK: Integrin-linked kinase, ADCC: Antibody-dependent cell-mediated cytotoxicity, HCC: Hepatoma carcinoma cell.

**Table 1 ijms-24-06170-t001:** Integrins, their ligands, and major functions.

Category	Integrin	Ligand	Major Function	Reference
RGD-binding integrins	αVβ1	TGF-β; fibronectin; osteopontin; neural cell-adhesion molecule L1	1. Inducing the accumulation of fibroblasts in ECM; 2. Inducing cytoskeleton reorganization; 3. Inducing cell differentiation; 4. Inducing vascular response to mechanical stimulation	[1,19,20,21,22,23]
	αVβ3	Vitronectin; TGF-β; fibronectin; osteopontin; neural cell adhesion molecule L1; fibrinogen; von Willebrand factor; thrombospondin; fibrillin; tenascin	1. Participating in angiogenesis, ECM regulation, vascular smooth muscle cell migration, and osteoclast adhesion to the bone matrix; 2. Regulating the migration and phagocytosis of immune cells	[1,24,25,26]
	αVβ5	TGF-β; osteopontin; vitronectin; bone sialic protein; thrombospondin; NOV	1. Participating in wound healing, matrix molecule synthesis, and type I procollagen expression; 2. Participating in cerebellar granule cell precursor differentiation	[1,27,28,29,30,31,32]
	αVβ6	TGF-β; fibronectin; osteopontin; ADAM	1. Regulating innate immunity and anti-inflammatory surveillance; 2. Participating in the process of tooth enamel formation	[1,33,34,35,36]
	αVβ8	TGF-β	1. Regulating neurovascular development; 2. Regulating immune cell recruitment and activation; 3. Regulating stem cell migration or differentiation	[37,38]
	α5β1	Fibronectin; fibrinogen; fibrillin; osteopontin; thrombospondin	1. Promoting cell migration, invasion, proliferation, and aging; 2. Maintaining the normal function of T cells	[39,40,41,42,43]
	α8β1	TGF-β; tenascin; fibronectin; osteopontin; vitronectin; nephronectin	1. A cell migration regulator	[1,44]
	αIIbβ3	Fibrinogen; fibronectin; thrombospondin; vitronectin; von Willebrand factor	1. Inducing platelet activation and arterial thrombosis	[1,45]
Leukocyte cell-adhesion integrins	αLβ2	ICAM-1	1. Regulating inflammation; 2. Inducing leukocyte spreading and crawling toward infection; 3. Enhancing the phagocytosis of neutrophils	[46,47,48]
	αMβ2	ICAM-2; C3d-opsonized particles; Zymosan; LL-37		[46,48,49,50,51,52]
	α4β1	VCAM-1	1. Slowing the rolling of leukocytes	[46]
	αXβ2	Osteopontin	1. Regulating the anti-inflammatory function of macrophages	[53,54,55]
	αDβ2	ICAM-1; ICAM-3; VCAM-1	1. Regulating inflammation	[48,56]
	αEβ7	E-cadherin	1. Mediating lymphocyte attachment to intestinal and skin epithelial cells	[57]
	α9β1		1. Regulating hematopoietic processes	[58,59]
	α4β7	VCAM-1; MAdCAM-1	1. Mediating the homing of lymphocytes to gut tissues	[60,61]
Collagen (GFOGER)-binding integrins	α1β1	Collagens I, III, IV, IX, XIII, XVI; Collagen IV chain-derived peptide	1. A promoter of T cells in inflammatory responses; 2. Mediating monocyte transmigration; 3. Participating in damage repair processes	[62,63,64,65,66]
	α2β1	Collagens I, III, IV, V, XI, XVI, XXIII; lumican; decorin	1. Stabilizing thrombi; 2. Mediating bone loss	[67,68,69,70]
	α10β1	Collagens II	1. Participating in cartilage production and skeletal development; 2. A biomarker of chondrogenic stem cells	[71,72]
	α11β1		1. Participating in tooth eruption; 2. Inducing osteogenic differentiation of mesenchymal stem cells; 3. Promoting myofibroblast differentiation; 4. Wound healing	[70,73,74]
Laminin-binding integrins	α3β1	Laminin-332; laminin-511	1. Mediating cell adhesion to the basement membrane and cell-to-cell communication	[75]
	α6β1	Laminin-111; laminin-511; laminin-332	1. Regulating luteal formation and follicle growth; 2. A regulator of angiogenesis	[76,77,78,79]
	α6β4	Laminins; epidermal integral ligand proteins	1. Maintaining the integrity and stability of epithelial cells; 2. Participating in the regulation of cell death, autophagy, angiogenesis, aging, and differentiation	[75,80,81]
	α7β1	Laminin-211; laminin-221	1. Participating in vascular development and integrity	[82,83]
	α1β1, α2β1, α10β1, αvβ3			[84,85,86]

RGD: Arg–Gly–Asp; ECM: extracellular matrix; NOV: nephroblastoma overexpressed; ADAM: a disintegrin and metalloproteinase; ICAM: intercellular adhesion molecule; VCAM: vascular cell-adhesion molecule; MAdCAM-1: mucosal address in cell-adhesion molecule-1.

**Table 2 ijms-24-06170-t002:** Integrin-targeted immunotherapies in clinical trials for cancer therapeutics.

Name	Targeting Integrins	Indication	Mechanism	Clinical Trial	Reference
CAR-T therapy	β7	Relapsed and/or refractory MS	Binding of CAR-T cells with integrins to enhance specificity		[224]
	αVβ6	Solid tumor			[222,223]
	αVβ3	Solid tumor			[1]
Oncolytic virotherapy	αVβ6	Pancreatic cancer and breast cancer	Modify specific virus to mediate immune killing		[228]
		Epithelial ovarian cancer			[229]
	β1	Solid tumor			[227]
ICI therapy	αVβ3	Solid tumor	Inhibition of the immune checkpoint on T cells to enhance T-cell immunotolerance		[202,232,233,234,235,236]
	αVβ6	Colorectal cancer			[211]
Other strategies	β4	B-CLL			[196]
	αVβ3	Solid tumor		NCT01806675 NCT00565721 NCT02775110	[232,233,234,237]
	α4β1	MS		NCT01440101 NCT00559702 NCT00744679 NCT02775110	[206,238,239,240,241]

CAR: Chimeric antigen receptor; ICI: Immune checkpoint inhibitor; MS: Multiple sclerosis.

## Data Availability

Not applicable.

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
