# Peer review of "The Interplay between Integrins and Immune Cells as a Regulator in Cancer Immunology"

_ijms, 2023, doi:10.3390/ijms24076170_

Round 1

Reviewer 2 Report

In the present review article manuscript, the authors discussed the two-edged role of integrins in the immunology and immunotherapy of cancers. The authors also highlighted some immunotherapies that target integrins which are in clinical trials stage. The topic is interesting and the manuscript is informative. References were properly cited in the text. 

There are few points to be considered before the manuscript is acceptable for publication:

1. It is recommended to provide an in-depth discussion of the immunotherapies under clinical trials, their limitations, and outcomes to inspire the clinical potential of these therapies.

2. Since the authors pointed to the necessity of recruiting improved experimental models, it is recommended to discuss the existing experimental models and their limitations.

3. The authors also pointed to integrins inhibitors as a promising future prospect. In such an area, it would be valuable to refer to gene therapies that target various integrins (e.g. siRNA).

4. The clinical trials.gov identifier for the clinical trials listed in Table 1 should be provided to facilitate easy access by the readers.

Reviewer 3 Report

The Review article authored by Quinfang Zhang et al aims to provide an overview of the interplay between integrins and immune cells and its biological significance in cancer.

After a short introduction concerning the structure and selectivity of integrins, the Authors address successively integrin biological activity in B cells, T cells, dendritic cells, natural killer cells, polymorphonuclear cells, and macrophages. Then, the manuscript describes the positive and negative impact on cancer growth, and the ongoing developments as therapeutic approach.

The topic is of importance and the manuscript would make therefore a suitable contribution for IJMS.

Nevertheless, the manuscript raises the following concerns

The Authors should provide a Table displaying the different integrin heterodimers (with the alternative names of the monomers), their selective ligands (matrix components; cell adhesion molecules; effectors e.g. TGFB, TH,…; …), which cell types express them, and the biological effects.

The authors might also propose a scheme of the different role of integrins in immune cells, e.g. adhesion on matrix/ migration, immune synapse, …

Line 84 Please specify “on postcapillary high endothelial venules in Peyer’s…”

Lines 195-197 Please rephrase, for instance such as “the binding of tenascin-C and osteopontin with α9 integrins on DCs and macrophages in cooperation with TLR signaling promote the production of IL-6 and IL-23, leading to the differentiation of naive CD4+ cells into Th17 cells and the development of arthritis”

Lines 100-101: "…to activate the AKT signaling pathway by which the phosphatidylinositol-3-kinase (PI3K) can be activated," In fact, this is the opposite, PIP3 generated by PI3K activate AKT.

Line 185 Please specify the interaction of integrin “with E-cadherin”

Lines185-186 This interaction does not trigger intestinal epithelial cell differentiation to “immune cells”

Lines 193-194 According to reference 152 ITGA4 is critical for Th1 T-cell infiltration in the brain, but not for Th17 T-cells

Lines 197-198 The Authors did not show that alpha5beta1 integrins “initiate leukemia”, but evidenced that CD154 inhibits apoptotic events in leukemia cell lines through the activation of prosurvival pathways, including p38, ERK1/2 and PI3K/Akt

Lines 412-414 Please change "to make contributes" to "to contribute", and after "Rac1/reactive" please add "oxygen"

Line 435 Please specify that alpha-v integrin acts as a receptor of TH

Lines 438-440 Please indicate which cell type (cancer cells ?) express alpha4 beta1 integrins

Lines 458-459 In my mind, if cancer cells lose their stemness, they will have a limited survival and would be sensitive to chemotherapy. This should not be considered as a positive effect on cancer development or detail this point

Lines 440-443 The meaning of this sentence is not clear. Please rephrase

Lines 480-481, Please change "reported to overexpress" to "reported to be overexpressed", and "which promotes" to "which promote"

The sentence lines 486-491 needs to be rephrased

Please check carefully all quoted papers in your manuscript. There is a serious problem with the references. This does not facilitate the reviewing ! Please, see below a list that is probably incomplete:

Line 87 ref 26 the 1st Author is Garçon (not Ballet as mentioned in the text), and his paper concerns T cells not B cells

Line 88 Ballet is quoted ref 24, not 26

Line 102 ref 28 is incorrect, it concerns dendritic cells

Ref 29 is a Review on PI3K signaling in cancer published in 2004, there are probably some more up-to-date reviews

Line 108 ref 31 is incorrect, it concerns DCs not B cells

Line 109 ref 32 is incorrect, it concerns neuroinflammation not IgA

Line 113 ref 33 is incorrect. It concerns alpha4 integrins in the homing of monocytes and DCs to gut and CNS

Line 121 ref 8 is incorrect

Line 143 the last name of the 1st Author from ref 38 is Bromley

Line 147 ref 40 concerns B cells not T cells, ref 41 concerns alpha v integrins in B cells not alpha 5 integrins in T cells. Reference 53 should be accurate.

Line 168 ref 37 concerns B cells not T cells

Line 195 the last name of the 1st Author from ref 9 is Kanayama

Line 198 the 1st Author of ref 53 is Bachsais, not Schippers

Line 205 the last name of the 1st Author from ref 54 is Braun

Line 440 ref 129 is incorrect

Line 502 "numerous studies" but only one reference (140) mentioned, that moreover is incorrect

Line 505 According to ref 132 and 143 it should be beta 6 not beta 8 integrin. For a role of beta 8 integrins in T-reg you might consider ref 144

Line 508 ref 142 is incorrect, it concerns NK cells

Line 513 ref 145 is incorrect, it concerns CAR-T cells

Line 531 the 1st Author is Mahiron, not Mahuron

Line 632 ref 169 is incorrect, it concerns the adverse effects of immune checkpoint inhibitors, but might be used to replace the ref in line 675

Reviewer 4 Report

The paper entitled "The Interplay between Integrins and Immune Cells as a Regulator in Cancer Immunology" is a well-organized review. English style and language are high-level, and the article is easy to read. I think it is a good quality review, but I have some little suggestions for the Authors, in order to further improve the quality of the manuscript:

1) on line 205 "Andrea et al." should be replaced with "Braun et al.", since Andrea is the first name of the author of the cited article.

2) on lines 53-54, the authors state: "However, dysregulation of integrins contributes to immune-related diseases such as multiple sclerosis (MS) and arthritis, or even cancer [8-10]". I suggest the authors to mention, besides MS, arthritis and cancer, some other diseases involving the interaction of immune cells integrins with their receptor: in particular, I suggest the authors to mention Alzheimer’s and Parkinson’s diseases (Glass et al., 2013), pulmonary pathologies (such as asthma and chronic obstructive pulmonary disease) (Lee and Yang, 2013), and atherosclerosis (Libby, 2002; Ailuno et al., 2021).

In this context I suggest the authors to add the following references:

- Libby, P., 2002. Inflammation in atherosclerosis. Nature, 420, pp.868–874

- Glass, C.K., Saijo, K., Winner, B., Marchetto, M.C. and Gage, F.H., 2010. Mechanisms underlying inflammation in neurodegeneration. Cell, 140, pp.918–934

- Lee, I.T. and Yang, C.M., 2013. Inflammatory signalings involved in airway and pulmonary diseases. Mediators of Inflammation, p.12, Article ID 791231.

3) The authors should improve the quality of the figures, in particular of figures 4 and 5, since the written parts are very difficult to read.

4) In section "3.3.4. Other strategies", I think the authors should add some reference to the strategy involving the use of integrin-simulating targeting molecules (in particular, targeting peptides), used to develop drug delivery systems targeting cancer cells; an example is the well-known VHPKQHR peptide, identified by Kelly et al. (2006) through a phage display technique, which is homologous to alpha-4 beta-1 integrin.

I suggest the authors to add the following references:

- K.A. Kelly, M. Nahrendorf, A.M. Yu, F. Reynolds, R. Weissleder, In vivo phage display selection yields atherosclerotic plaque targeted peptides for imaging, Mol. Imaging Biol. 8 (4) (Jul-Aug 2006) 201–207.

- G. Ailuno, S. Baldassari, G. Zuccari, M. Schlich, G. Caviglioli, Peptide-based nanosystems for vascular cell adhesion molecule-1 targeting: a real opportunity for therapeutic and diagnostic agents in inflammation associated disorders.  Journal of Drug Delivery Science and Technology 2020, 55.

- A. Koudrina, M.C. DeRosa, Advances in Medical Imaging: Aptamer- and Peptide-Targeted MRI and CT Contrast Agents. ACS Omega 2020, 5, 36, 22691–22701.

- S. Pastorino et al. Two Novel PET Radiopharmaceuticals for Endothelial Vascular Cell Adhesion Molecule-1 (VCAM-1) Targeting. Pharmaceutics 2021 Jul 6;13(7):1025. doi: 10.3390/pharmaceutics13071025.

Round 2

Reviewer 1 Report

The authors responded every comments. Publication is suggested. 

Reviewer 3 Report

The Authors have taken into consideration my comments. I considered that this revised version is acceptable for publication.

Due to time constraints, I did not check again the quoted references.

Last point, the Supplementary Tables should at least be mentioned in the text. Table 1 and Supplementary Table 1 could be merged in the manuscript, and Table 2 may deserve to be included in main text. Nevertheless, since these Supplementary Tables provide a series of references this may represent a source of error.

Sincerely yours
